# Qualitative and Quantitative Analytical Techniques of Nucleic Acid Modification Based on Mass Spectrometry for Biomarker Discovery

**DOI:** 10.3390/ijms25063383

**Published:** 2024-03-16

**Authors:** Ying Liu, Jia-Hui Dong, Xu-Yang Shen, Yi-Xuan Gu, Run-Hong Zhang, Ruo-Yao Cui, Ya-Hong Liu, Jiang Zhou, Ying-Lin Zhou, Xin-Xiang Zhang

**Affiliations:** 1Beijing National Laboratory for Molecular Sciences (BNLMS), Key Laboratory of Bioorganic Chemistry and Molecular Engineering of Ministry of Education, College of Chemistry and Molecular Engineering, Peking University, Beijing 100871, Chinazhouyl@pku.edu.cn (Y.-L.Z.); 2Analytical Instrumentation Center, Peking University, Beijing 100871, China

**Keywords:** nucleic acid modification, mass spectrometry, chromatographic separation, biomarker

## Abstract

Nucleic acid modifications play important roles in biological activities and disease occurrences, and have been considered as cancer biomarkers. Due to the relatively low amount of nucleic acid modifications in biological samples, it is necessary to develop sensitive and reliable qualitative and quantitative methods to reveal the content of any modifications. In this review, the key processes affecting the qualitative and quantitative analyses are discussed, such as sample digestion, nucleoside extraction, chemical labeling, chromatographic separation, mass spectrometry detection, and data processing. The improvement of the detection sensitivity and specificity of analytical methods based on mass spectrometry makes it possible to study low-abundance modifications and their biological functions. Some typical nucleic acid modifications and their potential as biomarkers are displayed, and efforts to improve diagnostic accuracy are discussed. Future perspectives are raised for this research field.

## 1. Introduction

Nucleic acid modifications play important roles in regulating gene expression, cell differentiation, and individual development [1,2]. These modifications do not change the gene sequence, but can expand genetic information [3]. Unlike gene sequences, these modifications dynamically change throughout an individual’s lifecycle thanks to various environmental factors. With the advance of research, more and more diseases have been proven to be related to changes caused by nucleic acid modification, and have been considered as potential targets for precise diagnosis and personalized treatment [4].

Commonly, a complete workflow for nucleic acid modification study can be divided into the following steps: (a) choose a biological model that may be influenced by nucleic acid modification; (b) determine the content of any nucleic acid modifications in the biological model using qualitative and quantitative strategies, and estimate the correlation between modified content and biological function; (c) screen target genes containing functional modification using sequencing technologies; (d) verify the functions of target genes in biological processes using molecular biology methods, and reveal the mechanisms of these functions; (e) verify the clinical performance and potential therapy among extensive clinical samples; (f) use the discovered nucleic acid modification as the biological model for diagnosis and therapy. It can be seen that the precise qualitative and quantitative analyses of nucleic acid modification provide almost the earliest direct evidence, and this result plays a strong guiding role for the subsequent research. Therefore, it is necessary to develop sensitive and reliable qualitative and quantitative methods.

Here, we will introduce DNA and RNA modifications, especially the well-known modifications. Then, we will introduce the detection strategies for nucleic modifications, focusing on mass spectrometry-based analytical methods. We will discuss the methods to enhance sensitivity based on appropriate enzymatic digestion and chemical labeling, as well as the latest research progress in disease diagnosis using nucleic modifications as biomarkers. We will also provide future perspectives for this research field.

## 2. Nucleic Acid Modifications

### 2.1. Modifications of DNA

As early as 1948, modified cytosine was first discovered on calf thymus DNA, and was inferred to be 5-methylcytosine (5mC) [5]. In 1975, researchers recognized that epigenetic information could be carried through chemical modification of cytosine [6,7]. As research continues, over 50 types of DNA modifications have been discovered in mammals, plants, and microorganisms [3,8]. The related chemical and biological properties of these DNA modifications are systematically organized in an open-source database, DNAmod (https://dnamod.hoffmanlab.org, accessed on 30 January 2024) [8]. After seven decades of development, the complete pathway of cytosine methylation and demethylation was discovered, including 5mC, 5-hydroxymethylcytosine (5hmC), 5-formylcytosine (5fC), 5-carboxylcytosine (5caC), and an abasic site (AP site) [9,10,11,12,13,14,15]. Other important DNA modifications include uracil (dU), 5-hydroxymethyluracil (5hmU), 5-formyluracil (5fU), *β*-D-glucosyl-5-hydroxymethyluracil (base J), *N*^6^-methyladenine (6mA), *N*^6^-hydroxymethyladenine (6hmA), *N*^6^-carbamoylmethyladenine (ncm6A), 2-aminoadenine (m2A), 8-oxo-7,8-dihydroguanine (8-oxo-G, OG), and 7-methylguanine (7mG) [3,8,16] (Figure 1A). Among them, the level of 5mC was reported as 2–7% of the genomic cytosine [17], while 5hmC was determined to be about 0.03–0.7% [15,18]. The contents were found to be 20 5fC and 3 5caC in every 10^6^ C, respectively, which nearly touches the sensitivity limit of direct mass spectrometry detection [10]. 5hmU, 5fU, and OG were found to be at either one or several bases per 10^6^ C level [19,20,21]. It can be seen that the discovery of modifications is related to the nucleic acid’s natural content, which indicates that improving the sensitivity of the method is of great importance for the discovery and research of new modifications [18].

### 2.2. Modifications of RNA

Compared to DNA modification, there are many more types of RNA modification. Over 170 RNA modifications have been found so far, which exist in almost all types of RNA, including messenger RNA (mRNA), transfer RNA (tRNA), ribosomal RNA (rRNA), and small nuclear RNA (snRNA) [22]. *N*^6^-methyladenine (m^6^A) is one of the most abundant RNA modifications in eukaryotes, with a content of 0.1–0.4% of total adenine, which is similar to pseudouridine (Ψ) content (0.2–0.6% of total uridine) [23,24]. Other important RNA modifications include *N*^6^-hydroxymethyladenine (hm^6^A), *N*^1^-methyladenosine (m^1^A), inosine (I), 2′-O-methyladenosine (Am), 5-methyluridine (m^5^U), 2′-O-methyluridine (Um), 5-methylcytosine (m^5^C), 5-hydroxymethylcytosine (hm^5^C), 3-methylcytosine (m^3^C), 2′-O-methylcytidine (Cm), 7-methylguanosine (m^7^G), and 2′-O-methylguanosine (Gm), et al. (Figure 1B). The related chemical and biological properties of these DNA modifications are systematically organized in an open-source database, MODOMICS (https://iimcb.genesilico.pl/modomics/, accessed on 30 January 2024), including LC–MS information, pathways, sequences, and related diseases [25,26].

## 3. Qualitative and Quantitative Analysis

The reported methods for nucleic acid modification quantification include thin-layer chromatography (TLC) [10], liquid chromatography (LC) or capillary electrophoresis (CE) based on optical detection [27,28,29], gas chromatography–mass spectrometry (GC–MS) [30], liquid chromatography–mass spectrometry (LC–MS), surface-enhanced Raman scattering (SERS) spectroscopy [31], immunoassay, and biosensing-based methods [32,33,34,35,36,37,38]. MS-based methods have been considered as the main analytical tool for nucleic acid modification quantification due to their wide applicability, excellent sensitivity, and wide linear range, providing comparative global compositional analysis of different biological samples. However, the low abundance of modifications limits the discovery of new modifications and the research of nucleic acid modifications as biomarkers or indicators. The main challenge is to establish simple, fast, and ultrasensitive global quantification methods to obtain more precise information for limited clinical samples. Effective sample preparation strategies, separation and detection processes, and data processing methods are important for accurate qualitative and high-sensitivity quantitative analysis, which help researchers to obtain reliable and comprehensive modification information (Figure 2).

### 3.1. Preparation of Biological Samples

#### 3.1.1. Hydrolysis

In mass spectrometry-based nucleic acid modification analysis, researchers first extract nucleic acids from biological samples, and use nucleases to hydrolyze the chain into individual nucleotides. Considering that the high hydrophilicity and negative charge of phosphate would decrease the ionization efficiency of mass spectrometry, phosphatase is used to remove phosphates and form deoxyribonucleosides or ribonucleosides. The classical digestion method developed by Crain and colleagues was divided into two steps: nuclease P1 or nuclease S1 first digests denatured DNA or RNA at 50 °C under pH 5 buffer, then the digestion solution is adjusted to pH 8, and phosphodiesterase and alkaline phosphatase are added to remove phosphates at 37 °C sequentially [39,40,41,42]. In this two-step method, nuclease P1 and nuclease S1 only recognize RNA or single-stranded DNA, and it is necessary to boil genomic DNA at 100 °C for denaturing first [43]. Quinlivan et al. developed a one-step digestion method, performed under an appropriate pH, by replacing nuclease P1 with endonuclease from *Serratia marcescens*, Benzonase, or DNase I, which has been widely commercialized owing to relatively fewer processing steps and a lower time consumption and dilution ratio [43,44]. In this one-step method, endonuclease from *Serratia marcescens* and Benzonase recognize single- and double-stranded DNA and RNA, and as the denaturing step is not essential, RNA samples could be hydrolyzed through the same workflow [43]. Directly hydrolyzing DNA or RNA into bases through heating under acidic conditions is also an alternative choice [45,46,47].

The aim of hydrolysis is to release nucleotides completely with an unbiased approach and reduce additional artificial modification and the loss of natural modification. The hydrolysis method should be carefully chosen for certain modifications, especially for low-abundance modifications, which might cause more bias in quantification results. Weinfeld et al. reported that several enzymes worked through stacking with aromatic nucleic acid bases [48]. Yuan et al. first reported the resistance of 5caC to phosphodiesterase I (PDE1), and found that commercial one-step digestion mix was suitable for 5caC hydrolysis [49]. Chu et al. reported that PDE1 released m^7^G fully, both in mRNA sequence and in the 5′ cap; contrarily, S1 nuclease released the internal m^7^G with high activity but released m^7^G from the 5′ cap with much lower activity [50]. Different digestion methods were recommended for studies of 2′-*O*-methylated ribonucleosides [51], RNA 5′ caps [52], et al. Besides the base structure, sequence specificity also needs to be considered; for instance, the formation of G-quadruplex (G4) inhibits the cleavage efficiency of nuclease [53,54]. Caution also needs to be used regarding digestion conditions during special modification quantification. Matuszewski et al. reported that hydantoin *N*^6^-threonylcarbamoyladenosine (ct^6^A) was converted to stereoisomer under mild alkaline conditions in several minutes [55]. The extreme pH, radical species, buffer types, and deaminase contamination may cause changes in modification type and intensity, including A to I, G to OG, m^1^A to m^6^A, and 5-methoxycarbonylmethyl-2-thiouridine (mcm^5^s^2^U) to 5-methoxycarbonylmethylisocytidine (mcm^5^isoC) artifacts, which can be avoided by adding metal-chelating reagent, antioxidant, or deaminase inhibitor and treating in mild conditions [40,44,56,57,58].

In addition, a simpler workflow and shorter preparation time are also desirable goals. Lai et al. reported that appropriate concentration of divalent Mg^2+^ and Ca^2+^ promoted DNA digestion catalyzed by the DNase set, but high concentration of monovalent Na^+^ and K^+^ inhibited DNA digestion [59]. Engineered nuclease mutant was produced for the hyperactive and unbiased release of DNA modifications, which better overcame the inhibition of buffer conditions [60,61,62]. To shorten the preparation time, Yin et al. designed a cascade bioreactor by immobilizing nucleases on a capillary silica monolith, and DNA was completely digested within 10 min [63].

#### 3.1.2. Nucleoside Extraction

The purpose of nucleoside extraction is to improve separation and detection efficiency and obtain higher signals of the analytes by removing salts, proteins, and uninterested nucleosides, especially large amounts of normal nucleosides, from the matrix. The initial extraction methods enrich free nucleosides from biological samples such as serum and urine, normally based on hydrophobic or hydrophilic interactions, ion exchange, and affinity with 1,2-cis-diol compounds with commercial materials, such as HLB [64,65] and WCX [66], as well as new materials such as graphene [67], polymers [68], metal oxides [69,70,71], and boronate-decorated substrates [72,73,74], which are also used for the extraction of nucleoside products after enzymatic digestion to further increase signal response. The shapes of substrate and extraction devices influence extraction efficiency [75,76]. However, most of the above methods either lack selectivity for specific types of nucleoside modification or discriminate against DNA nucleosides and 2′-*O*-methylated ribonucleosides.

In order to extract specific structural nucleosides, various novel functional materials have been designed and synthesized. Ma et al. developed a pH-response covalent organic framework decorated with gold nanoparticles as a linker and glutathione as a functional group, which selectively captured m^1^A via electrostatic interaction and released it in an acidic condition to prevent m^1^A’s rearrangement to m^6^A [77]. Wang et al. developed cyclodextrin-based porous liquids, which showed a chiral recognition and separation ability of _D_-type and _L_-type pyrimidine nucleosides [78].

#### 3.1.3. Chemical Labeling

The quantification of ultra-rare nucleic acid modifications always needs a large amount of sample to gain enough response from detectors. In contrast, the capacity of chromatographic columns and the linear range of mass spectrometry seem to be limited to the direct quantification of multiple nucleosides with large differences in content. Moreover, limited biological and clinical samples mean that direct detection is far from meeting the requirements of real research. Therefore, chemists have developed brilliant labeling strategies based on chemical and enzymatic catalyzed reactions to improve detection sensitivity. Table 1 summarizes the application of labeling reagents in modified nucleoside detection.

In one of the strategies, labeling reagents react indiscriminately with all nucleosides. Pertrimethylsilylation, permethylation and *O*-isopropylidenation are achieved by adding *N*-methyl-*N*-(trimethylsilyl)trifluoroacetamide (MSTFA), bis(trimethylsilyl)trifluoroacetamide (BSTFA), iodomethane, and acetone. These methods are usually simple, robust, and low-cost, and are mainly followed by GC–MS detection [30,71,79,80,81]. 8-(diazomethyl)quinoline (8-DMQ), dimethyl-*p*-phenylenediamine (DMPA), and 2-(diazomethyl)-*N*-methyl-*N*-phenyl-benzamide (2-DMBA) are synthesized and used to react with phosphate groups to enhance the mass spectrometry response of nucleoside triphosphates and nucleotides extracted from endogenous metabolites [82,83,84].

In other designs, labeling reagents only react with specific modified nucleosides. α-haloketones, including 2-bromo-1-(3,4-dimeth oxyphenyl)-ethanone (BDMOPE), 2-bromo-1-(4-methoxyphenyl)-ethanone (BMOPE), 2-bromo-1-(4-diethylaminophenyl)-ethanone (BDEPE), and 2-bromo-1-(4-dimethylamino-phenyl)-ethanone (BDAPE), react with the N3 and N4 positions of cytosine to form the cyclic derivatives [85,86]. Several more bases other than cytosine were found to be derived from α-haloketone reagent [87]. Hydrazine group-based reagents display high reactivity and selectivity to react with different modified nucleosides under different conditions. Yu et al. and Yuan et al. developed a series of reaction routes through labeling the aldehyde group with hydrazino-*s*-triazine-based reagents (Me_2_N, Et_2_N, and i-Pr_2_N) [88,89] directly, labeling the carboxyl group under catalysis, and converting the hydroxyl group to the reactive aldehyde group through oxidation before labeling (5hmC to 5fC, and 5hmU to 5fU), which increased the sensitivity of 5fC, 5caC, 5hmC, 5fU, and 5hmU by up to 850 folds (Figure 3) [90,91,92]. Other reported hydrazine-based labeling reagents include Girard’s reagents (GirP, GirT, GirD and 4-APC) [93,94] and rhodamine B hydrazine [95]. In addition, cationic xylyl-bromide (CAX-B) [96], *N*-dimethyl-amino naphthalene-1-sulfonyl chloride (Dns-Cl, Dens-Cl) [97], hydroxyl amine-based reagents [98], and *N*-cyclohexyl-*N*′-*β*-(4-methylmorpholinium) ethylcarbodiimide *p*-toluenesulfonate (CMCT) [99] play important roles in labeling and improving the sensitivity of nucleic acid modifications. Enzyme-based methods possess good specificity of recognition and enrichment, although they may change the original structure of the nucleosides. Tang et al. reported a method which covalently added a glucosyl group to 5-hmC using T4 β-glucosyltransferase and enriched the product by hydrophilic interaction [100]. Hu et al. reported an enrichment method for 5′ NAD^+^ modification, in which the NAD^+^ group was replaced with a click reaction group through the catalysis of adenosine diphosphate ribosylcyclase (ADPRC), and the product was enriched for quantification and sequencing [101].

Some general characteristics of well-designed labeling reagents include an efficient and selective reaction group, a hydrophobic backbone group, and an easily charged tertiary or quaternary amine group. These function groups block the negatively charged and hydrophilic structure, enhance the organic solvent proportion during LC, and make the targets easy to distribute on the surfaces of droplets during ESI, resulting in the increase of spraying ability, ionization efficiency, and mass spectrometry response. In addition to ultrasensitive determination, labeling reaction is also important for relative quantification and structural identification. Several reagents with stable isotope labeling or similar structures were synthesized and applied in pairs [84,97].

**Table 1 ijms-25-03383-t001:** Chemical labeling strategies in high-sensitivity detection of nucleic acid modifications.

Labeling Reagent	Structure	TargetNucleoside	Reaction Condition	LOD	SensitivityIncrease Fold	SampleConsumption	Ref.
MSTFA	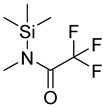	all	10 μL of methoxyamine hydrochloride as 20 mg/mL solution in pyridine, 90 μL of MSTFA	ND.	ND.	metabolites in 25 μL blood plasma, 5 × 10^6^ cells, or 5 mg tissues	[30]
acetone	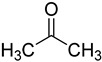	ribonucleosides	400 μL of acetone with *p*-toluene sulfonic acid (1 mg/mL), 50 °C, 2 h	0.6–6.5 fmol	7–30 folds	metabolites in 100 μL of urine	[71]
BSTFA	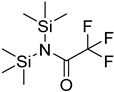	all	120 μL of BSTFA, 70 °C, 2 h	ND.	ND.	10 mg of freeze-dried leaves	[79]
acetone	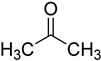	ribonucleosides	600 μL of acetone, 6 μL of HClO_4_, vortex for 30 s, −20 °C for 30 min	0.026–0.16 ng/mL, 10 μL	ND.	metabolites in 100 μL urine	[80]
iodomethane-d3	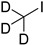	all	iodomethane-d3, on beads, room temperature, 10 min	10 fmol/μL, 2 μL	ND.	1 µg of purified DNA or RNA	[81]
8-DMQ	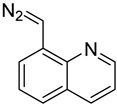	nucleoside triphosphates	50/1 molar ratio of 8-DMQ/analyte in 50 mM borate buffer (pH 6.9, 160 μL) with DMSO (40 μL), 25 °C,10 min	0.4–1.3 fmol	56–137 folds	metabolites in 1.0 × 10^7^ cells	[82]
DMPA	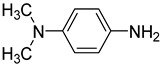	nucleotide	molar ratios of DMPA and EDC over nucleotides were set as 40,000 and 5000, with 100 μL of imidazole solution (1 mM, pH 6.0), 50 °C, 1.5 h	0.12–0.47 fmol	88–372 folds	metabolites in urine, tissue and cell line samples	[83]
2-DMBA, d5-2DMBA	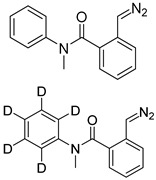	nucleotides, nucleoside diphosphates, nucleoside triphosphates	200 μL of 250 mg/L 2-DMBA in pH 7.0 borate buffer, 30 °C, 30 min	0.07–0.39 fmol	17–174 folds	metabolites in 20 mg of tissue and cell line samples	[84]
BDAPE	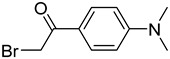	5mC, 5hmC, 5fC, 5caC	4 mM of BDAPE in 200 μL of ACN using 4 mM Et_3_N as the catalyst, 60 °C, 6 h	0.06–0.23 fmol	35–123 folds	10 μg of genomic DNA	[85]
BDMOPE, BMOPE, BDEPE	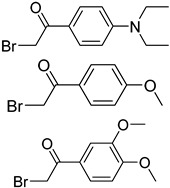	m^5^Cm, hm^5^Cm, f^5^Cm, ca^5^Cm	6 mM BDMOPE and 6 mM triethylamine, 60 °C, 6 h	0.06–0.22 fmol by BDMOPE labeling	46–462 folds	10 μg of total RNA and small RNA	[86]
BrDPE	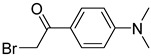	C, dC, A, dA, G, dG, T, dT, U	BrDPE/analyte ratio 200/1 and 4 mM triethylamine in 125 μL solvent, 40 °C, 3 h	0.3–12.5 fmol	31–107 folds	metabolites in 0.2 g of dry sample	[87]
Me_2_N, Et_2_N, and i-Pr_2_N	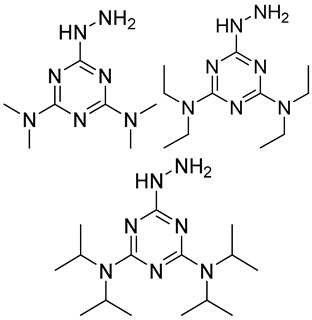	5fC, 5caC	5 mM labeling reagents with 1% HAc in 20% MeOH votex for 10 s for 5fC; labeling reagents in 20 μL 50% ACN, 10 μL 4 mg/mL HOBT and 10 μL 50 mg/mL EDC, 37 °C, 30 min for 5caC	10–25 amol	100–125 folds	600 ng of genomic DNA	[90]
i-Pr_2_N	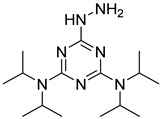	5hmC	5 mg of MnO_2_ in 20 μL reaction volume, 50 °C, 1 h; 5 μL of oxidation product, 1 μL of HAc, 14 μL of 50 mM i-Pr_2_N solution, vortex and dry	14 amol	178 folds	0.6–2.4 ng of cell-free DNA	[91]
i-Pr_2_N	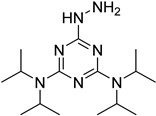	5fU, 5hmU, 5fC, 5hmC	5 mg of MnO_2_, 2 μL FA, 50 °C, 1 h; 1 mg/mL i-Pr_2_N and 2 µL HAc, vortex	26.0–44.4 amol	275–850 folds	2 μg of genomic DNA	[92]
GirP, GirT and 4-APC	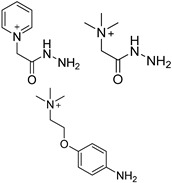	5fdC, 5frC, 5fdU, 5frU, 5frCm, 5frUm	GirP/analyte ratio 50/1, 30 °C, 5 min	0.03–0.05 fmol	115–880 folds	mixture of 10 μg genomic DNA and 10 μg total RNA	[93]
GirP, GirT and GirD	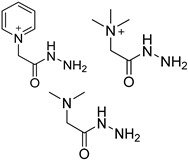	5fC, 5caC	GirD/analyte ratio 50/1–150/1, 40 °C, 5–40 min	0.03–0.42 fmol	52–260 folds	20 μg of genomic DNA	[94]
rhodamine B hydrazine	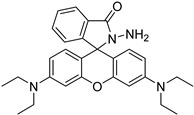	5fC	10 μL of 5 mM labeling reagent in MeOH, 0.2 μL HAc, vortex and dry	3 amol	300 folds	total RNA in cell line sample	[95]
CAX-B	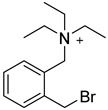	bases that have active hydrogen	CAX-B in 50% ACN (20 mg/mL), with Et_3_N(20 μL/mL), was mixed 1:1 with the sample solution, 45 °C, 2 h	160 amol thymidine	ND.	ND.	[96]
Dns-Cl, Dens-Cl	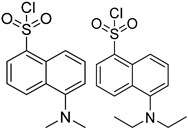	C, dC, 5mdC, m^5^C, A, m^1^A, m^6^A	100 μL of reaction buffer (pH 11) and 100 μL of Dns-Cl, 30 °C, 1 h	0.001–0.01 μg/mL, 5 μL	1.6–400 folds	metabolites in 10^6^ cells	[97]
hydroxyl amine	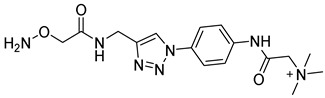	AP sites, βE sites	1.5 mM derivatization reagents in HEPES (20 mM, pH = 7.5) and Na_2_EDTA (0.1 mM), 37 °C, 40 min	0.11 fmol	ND.	5–20 μg of genomic DNA	[98]
CMCT	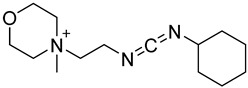	Ψ, U, m^5^U, m^6^U, mcm^5^U, hm^5^U, m^1^Ψ, mo^5^U	50 mM CMCT in borate buffer (50 mM, pH 8.5), 40 °C, 14 h	0.29–2.20 fmol	6–1408 folds	500 ng of mRNA	[99]

ND, not determined. ACN, acetonitrile; βE sites, β-elimination sites; EDC, N-(3-(Dimethylamino)propyl)-N′-ethylcarbodiimide hydrochloride; Et_3_N, trimethylamine; FA, formic acid; HAc, acetic acid; HEPES, 2-[4-(2-hydroxyethyl)piperazin-1-yl]ethanesulfonic acid; HOBT, 1-hydroxybenzotriazole hydrate.

### 3.2. Chromatography-Coupled Mass Spectrometry Technique

#### 3.2.1. LC–MS

Compared with nucleic acid determination based on immunoassay and optical detection, the liquid chromatography–mass spectrometry-based method provides better accuracy for qualification and higher sensitivity for quantification, better robustness for different sources of samples, and richer information regarding the discovery of unknown modifications, which is why it has become the most widely used method currently. The first discovery of 5hmC in mammalian cells was achieved using HPLC-MS in 2009 [9,15], and further studies have been performed continuously over the past decade, including the identification of new modifications [10,14,102]. In order to improve the detection efficiency, researchers focused on the improvement of the mass spectrometry mode, ionization mode, separation mode, and buffer addition, reducing the sample consumption to limited numbers of cells or even single cells.

With the development of mass spectrometry technology, the multiple reaction monitoring (MRM) mode of tandem mass spectrometry (MS/MS) was used for nucleic acid modification detection, and the limit of quantification (LOQ) of 5mC was 40 fmol in the report [41]. Targets were selected by precursor ions and fragment ions stepwise in the MRM mode, which improved the accuracy of structural identification and reduced the background, effectively improving the sensitivity. Multistage MS (MS/MS/MS) was introduced during modification identification to supplement MS/MS, especially for isomers which share precursor ions and fragment ions, such as m^3^U and m^5^U, or m^1^A and m^6^A [103,104]. The process of nucleoside electrospray ionization (ESI) was estimated, and the sensitivity was found to be limited by the formation of dimers, ion adducts, and in-source fragmentation, which could be prevented through the optimization of pipeline material, gas flow, heating temperature, collision energy, LC elution buffer, and appropriate derivatization reagent [95,105].

Since the development and commercialization of nanoliquid chromatography (nanoLC) and nanospray ESI, an extremely low flow rate of ~20 nL/min has been obtained, which helps to enhance desolvation, ionization efficiency, and the tolerance to salt, and greatly improves the sensitivity of nucleic acids [106,107,108]. Two-dimensional nanoLC involves a pre-column before the separation column. The nucleosides are trapped and concentrated on the pre-column, and then they enter the analysis column for separation, which is caused by gradually changing the elution solution [103,109,110,111]. The column stationary phases are also critical to separation [112], and are often replaced with novel materials that interact with nucleosides for online enrichment and improvement of separation [73,113].

Reasonable selection of the chromatographic mode can improve resolution and sensitivity, and reduce the signal overlap. The earliest research used reverse phase liquid chromatography (RP-LC) to separate nucleosides based on the C_18_ stationary phase [41]. Amide, perfluorinated phenyl (F5), and T3 bonding analytical chromatography columns were compared for chromatography separation [114]. Researchers compared different types of reverse phase stationary phases and introduced hydrophilic interaction liquid chromatography (HILIC), considering the relatively high hydrophilicity of most nucleosides [47,115]. The interaction between the target and the stationary phase and the detection sensitivity can be improved by adding formic acid [105], ammonium salts [64], nonafluoropentanoic acid [116], and malic acid [67,117] to the mobile phase.

#### 3.2.2. CE–MS

Capillary electrophoresis is a separation technique driven by a high-voltage electric field in a capillary, which separates charged particles based on their mobility. It has higher separation efficiency, faster analysis speed, and lower sample consumption compared to liquid chromatography, and is especially suitable for the separation of polar nucleoside molecules. Early researchers used CE–UV to achieve the separation and detection of C and 5mC within 1.5 min [118,119]. Considering that the sensitivity of UV detectors was relatively low, fluorescence labeling methods were developed for highly sensitive detection using laser-induced fluorescence (LIF) detectors [120]. 

With the interface technology of capillary electrophoresis–mass spectrometry (CE–MS) becoming more mature [121,122,123,124,125,126], stability and sensitivity has become good enough to be applied to modified nucleoside detection [127,128]. Yuan et al. and Yu et al. developed ultrasensitive and simultaneous determination methods for DNA and RNA modified nucleosides. A sheathless interface prevented the dilution effect in these reports, a limit of detection (LOD) down to 2.5 amol was achieved, and sample consumption was reduced to a limited number of cells [129,130]. Lechner et al. analyzed RNA modifications at both the nucleoside level and the oligonucleotide level by using CE–MS, reached 68–97% sequencing coverage of the RNA, and separation of four methylated guanosine isomers (1-methylguanosine; N2-methylguanosine; Gm and m^7^G) was performed [131].

#### 3.2.3. Other Mass Spectrometry-Based Techniques

Besides ESI and nanoESI introduced by LC and CE, electron ionization (EI) and chemical ionization (CI) combined with GC were also widely used [79,132,133], although reports suggested that high temperatures might cause changes to the target structure [134]. Direct injection [109] and ambient ionization, such as direct analysis in real time (DART) [135,136], create advantages for analysis speed. Ion mobility spectrometry (IMS) separates ions based on collision cross-sections in the millisecond time scale, and IMS has a sensitivity of 15 pmol of adenosine, displaying a potential ability to separate isomeric nucleotide and nucleoside variants in complex samples based on subtle differences in ion mobility behaviors, which is another dimension in addition to mass-to-charge ratios [137,138,139,140].

## 4. Data Analysis

Effective data processing helps to identify the structure of the analytes accurately, quantify the content of the analytes precisely, and discover unknown modifications, and it makes simultaneous and automated analyses possible. 

In order to determine the analytes and avoid the interference from similar molecular weight substances and isomers, the following methods are usually used: (a) Use high-quality resolution mass spectrometry. Accurate mass-to-charge ratio excludes other types of molecules with similar molecular weights [109]. (b) Analyze the structure of secondary fragments. Isomers usually produce different fragment ions through tandem mass spectrometry, and multistage mass spectrometry is also used [136]. The structural analysis of fragments is also performed by comparing between naturally modified nucleosides and stable isotope-labeled nucleosides [141]. Chen et al. discovered that in-source fragmentation usually occurs in glycosidic bonds, and they found a correlation between glycosidic bond length and cleavage ratio through theoretical calculations, and proposed a qualitative method based on the mass spectrometry of parent ions and in-source cleavage fragments [95]. (c) Compare chromatographic retention times. Gonzalez et al. found that nucleoside retention times of chromatography follow a certain regular pattern according to the hydrophilicity, hydrophobicity, and other properties of nucleosides, which promotes the accuracy of qualification [104].

In order to process data in batches automatically for discovering and quantifying modified nucleosides, several types of software have been developed and applied. Commercial software, such as Compound Discoverer 3.0, is used to search for modified nucleosides using metabolite analysis workflows [142]. Specialized software, such as Nucleos’ID (https://github.com/MSARN/NucleosID, accessed on 30 January 2024) and NuMo Finder (https://github.com/ChenfengZhao/NuMoFinder, accessed on 30 January 2024), has been developed based on the retention time of liquid chromatography and capillary electrophoresis, as well as the mass of precursor ions and fragment ions provided in the database [81,143]. Nucleosides are classified into networks based on fragments through the establishment of a mass spectrometry database, which contributes to the discovery of unknown modifications [144]. These schemes are also used as part of mass spectrometry-based oligonucleotide sequencing [145,146]. The development of artificial intelligence (AI) brings more potential tools for nucleic acid modification data analysis [147]. The preparation and updating of databases are of great importance for automated processing engines, and MODOMICS and DNAmod are the most commonly used databases that contain nucleoside mass spectrometry information as mentioned above.

## 5. Disease Diagnoses Based on Nucleic Acid Modifications

Numerous studies show that nucleic acid modification is widely involved in important biological processes such as embryonic development, cell differentiation, and life rhythm by regulating gene transcription and expression. Abnormal nucleic acid modification is found to be associated with cancer [148], nervous system diseases [149], immune diseases [150], diabetes [151], and other diseases. Nucleic acid modification responds quickly to environmental stress and occurs in the early stages of most tumors, even before some oncogenic gene mutations occur, and is considered as an ideal biomarker for early diagnosis of cancer [152].

Thanks to the earliest studied modification of DNA, 5mC, researchers have reached a consensus on cancer biomarkers, which is that the methylation level of the entire genome of cancer cells is lower than that of normal cells, while the methylation level of specific genes is higher [4]. Various modifications, such as 5hmC of DNA, and m^6^A, Ψ, and m^5^C of RNA, have also been found to be associated with cancer, and their regulation mechanisms are still being studied [13,153,154]. In recent years, You et al. identified a new adenosine dual methylation modification, m^1,6^A, in mammalian cells, which has an abnormal upward trend in breast cancer tissue [155]. The discovery of these patterns of change and biological mechanisms has led nucleic acid modification to be increasingly regarded as a disease biomarker. 

In addition to the application of a single biomarker, accumulating evidence suggests the advantage of combing of multiple biomarkers. Yu et al. developed an ultrahigh-sensitivity method for detecting multiple important modifications in cancer and adjacent tissues simultaneously, and found that the detection rates of 5hmC, 5hmU, and 5fU alone as a biomarker for breast cancer samples were 95%, 75%, and 85%, respectively; while by detecting these three cancer biomarkers simultaneously, two of the three were 100% consistent with the overall trend [92]. Therefore, simultaneous detection of multiple nucleic acid modifications as cancer biomarkers in clinical samples greatly improved the accuracy of cancer diagnosis. Tian et al. found the crosstalk between DNA 5mC and RNA m^6^A in hepatocellular carcinoma, and developed an epigenetic and epitranscriptomic module eigengene (EME) to optimize risk stratification better and predict the clinical outcomes and progression of patients [156].

The selection of biological tissues is important for modified nucleosides as tumor markers. The techniques for separating and preparing tissues into subdivided samples, such as cancer and adjacent tissues [92], circulating tumor cells [157], and cell-free DNA in blood [91], exosomes [158], and even single cells [159] have been widely studied, and the differences in modifications between different samples have been discovered. Yokoi et al. found that cancer cells secrete more exosomes that contain genomic DNA than normal cells [160], and Pan et al. found that the content of m^6^A in cancer-derived exosomal small RNAs is higher than that in the cells found by LC–MS/MS systematic profiling [158]. In addition, liquid biopsy could be achieved through separating cell-free DNA and exosomes, which has significant advantages, such as a simple and convenient sampling operation; low cost; more flexible and safe collection of samples throughout the entire disease process; dynamic monitoring of tumor progression and genetic changes; and detection of circulating biomarker targets generated in the early stages of cancer, achieving early diagnosis of cancer [161,162].

## 6. Conclusions and Perspectives

In conclusion, nucleic acid modification plays an important role in biological activity and disease occurrence. Exploring the relationship between modifications and diseases, searching for new biomarkers, and developing more accurate detection methods for diagnosis are urgent clinical needs. The overall quantification based on mass spectrometry can quickly obtain information about modification types and intensity, guiding research directions. Sensitive, fast, simple, and low-cost detection methods were reported for different specific applications by developing appropriate sample digestion, derivatization, and separation detection schemes.

In the future, more sensitive and selective quantification methods will still be one of the research focuses, promoting more precise zoning detection and single-cell detection to achieve accurate diagnoses. In addition, the unknown modifications are mainly confirmed through accidental discovery and chemical synthesis verification [57,155]. It is expected that the discovery of new modifications will be achieved through higher-sensitivity detection and better untargeted tandem mass spectrometry data processing methods. Meanwhile, quantification methods based on chemical labeling can be extended to the development of sequencing methods [49,163], obtaining information about the modifications to the genome and transcriptome, and investigating the mechanisms by which modifications affect biological functions. With more biological models validated, more analytical methods will be applied to life processes and diseases that are currently difficult to explore.

## Figures and Tables

**Figure 1 ijms-25-03383-f001:**
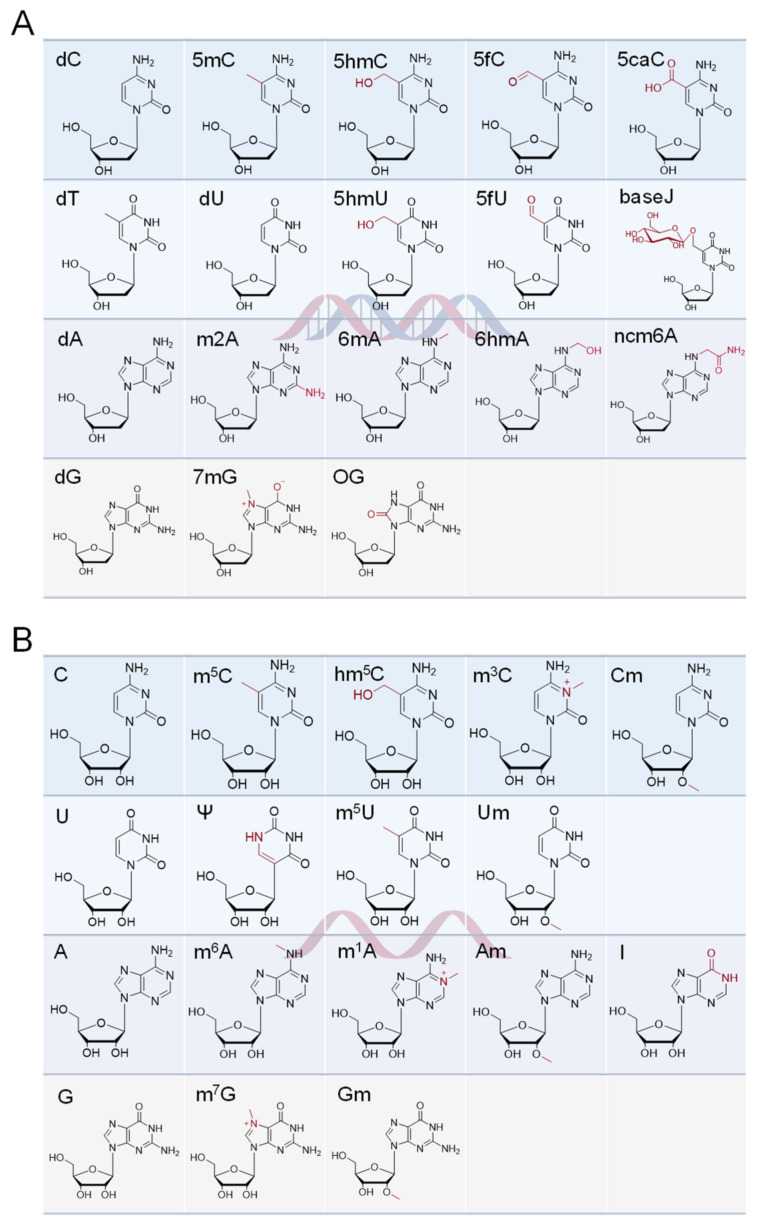
The natural nucleosides and some of the most studied modified versions found in (**A**) DNA and (**B**) RNA.

**Figure 2 ijms-25-03383-f002:**
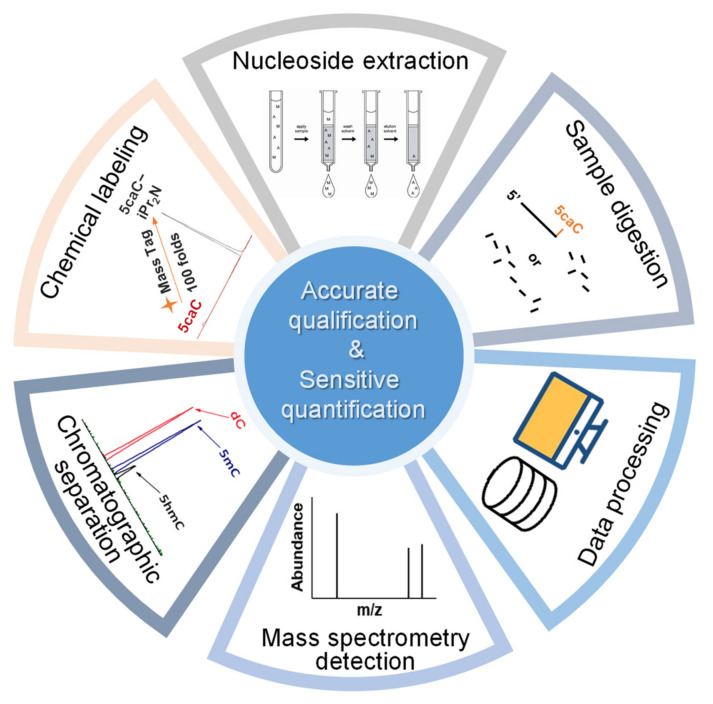
Process of mass spectrometry-based method for accurate qualification and sensitive quantification of nucleic acid modification.

**Figure 3 ijms-25-03383-f003:**
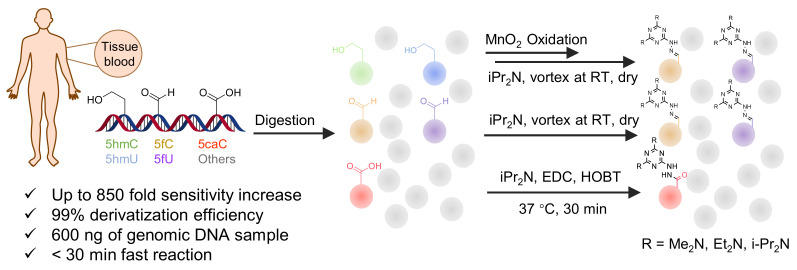
Workflows of derivatization of five types of DNA modification (5hmC, 5fC, 5caC, 5hmU, and 5fU) by hydrazino-*s*-triazine-based reagents.

## Data Availability

No new data were created or analyzed in this literature review. Data sharing is not applicable to this article.

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
