# Peer review of "Qualitative and Quantitative Analytical Techniques of Nucleic Acid Modification Based on Mass Spectrometry for Biomarker Discovery"

_ijms, 2024, doi:10.3390/ijms25063383_

Round 1

Reviewer 1 Report

Comments and Suggestions for Authors

The manuscript «Qualitative and quantitative analytical techniques of nucleic acid modification based on mass spectrometry for biomarker discovery» submitted by Ying Liu et al. to International Journal of Molecular Sciences is devoted to the review of approach to analysis of nucleic acid modification using mass spectrometry. Investigation of qualitative and quantitative composition od nucleic acid modification can be fruitful for development of new approach for diagnosis of different diseases such as cancer. The main problem of such studies is low content of nucleic acid modifications in biological samples. Authors discussed the variants of improvement of detection sensitivity and specificity and future perspectives of this direction.

The review is comprehensive and will be interesting for researcher from diverse scientific areas.

I have some questions and remarks:

1)      The Figure 1A must be mentioned in the Text as the Figure 1B. In present variant of manuscript the description of Figure start from B panel.

2)      The structures on the Figure 1 are situated little bey disorderly. Please change the Figure to add some order to the structures.

3)      Some pictures on Figure 2 are not visible. Please enhance the visibility of pictures Chromatographic separation and Chemical labeling. Is not clear the meaning of Chemical labeling picture. What does it mean? Please change the picture of add the capture.

4)      Section 3.1.1 Hydrolysis. Could you please separate DNA and RNA hydrolysis? Are there any differences in hydrolysis methods for these two types of nucleic acids?

5)       Section 3.1.2 Nucleoside extraction. In publication 79 and 80 authors used the modification. I recommend to move this work at section 3.1.3. Chemical labeling.

6)      Figure 3. The colors of circles are changed after chemical reactions. Does it mean the changing of nucleoside structure or all changes are indicated by chemical formulas?

7)      Abbreviation LOD (line 292) is not decoded in the text. Please add the decoding.

8)      There are comparatively lot of mistakes and mistyping in the text. Please check the text.

Comments on the Quality of English Language

There are comparatively lot of mistakes and mistyping in the text. Moderate editing of English language required 

Author Response

Dear editor,

Thank you for your letter and for the reviewers’ comments concerning our manuscript (ijms-2873435) entitled “Qualitative and quantitative analytical techniques of nucleic acid modification based on mass spectrometry for biomarker discovery". Those comments are all valuable and helpful for revising and improving our paper. The main corrections and the responds to the reviewers’ comments are listed point by point in the attached file.

I look forward to hearing from you.

Best regards,

Dr. Xin-Xiang Zhang

College of Chemistry and Molecular Engineering

Peking University

Reviewer 2 Report

Comments and Suggestions for Authors

The manuscript reviews the analytical methods based on mass spectrometry used for the determination of modified nucleosides and nucleotides in biological samples. The manuscript is interesting, well-organized, and figures are illustrative. References are mostly published recently and it is informative for a broad audience. English is mostly correct but still contains several grammatical mistakes. I will include a few of them at the end of this comments but, it may be there are more to be spotted.  In my opinion the manuscript can be accepted after careful English correction.

Page 1:

Abstract: Some of the phrases are too long. In addition in line 16 it says “he” and it should be “the”; line 19 it says “analysis” and it should be “analytical”; line 21 it says “diagnose” and it should be “diagnostic”.

Introduction: Line 27 at the beginning of the introduction, the expression “in the genome of life” is strange and it may not be accurate as some of the nucleic acid modifications are not in the genome but they can be the result of posttranscriptional modifications. I believe this expression can be removed without losing the meaning of the first phrase of the introduction.

Page 2: lines 47-49. The whole phrase “Here, we introduce the typical known modifications of DNA…… mainly focusing on mass spectrometry  based analysis methods” is difficult to read and the expression “typical known modifications” is not correct. I believe authors mean “well-known modifications”, or “most important modifications”, or “more abundant modifications” or “more studied”. I will suggest: “Here, we introduce the detection strategies for DNA and RNA modifications especially for known nucleic acid epigenetic modifications,  focusing on mass spectrometry based analytical methods.”

Page 3:

Figure 1. I will change the figure legend from “Typical modification types in (A) DNA and (B) RNA” for “Natural and some of the most studied modified nucleosides found in (A) DNA and (B) RNA. This is because the figure contains the structures of the natural nucleosides, so a natural nucleoside cannot be considered a typical modification type. In addition I will suggest using the abbreviation dA, dG, dU and dC for the 2’-deoxy nucleosides because one single abbreviation (A, C, U, G) cannot be used to describe two different compounds.  In addition the formulae of dU, 5hmU and 5fU are incorrect as they are not cyclopentane derivatives. It lacks the oxygen in the deoxyribose unit.

Page 4, line 124, it says “Benznase” and it should be “Benzonase”

Page 5, line 149, it says “the goals that researchers concern about”, I recommend “the desirable goals”; line 155, it says “necleases” and it should be “nucleases”.

Pages 7 and 8, Table 1. Authors say that the abbreviation “ND” means “not detected”. I believe they meant “not determined” because these data are not reported. If the labeling is not detected then it should not be in the table.

Page 9, line 243 it says “in mammal cell” and it should be “in mammalian cells”

Page 11, line 342 it says “Disease diagnose” and it should be “Disease diagnostic”; line 380 it says “exsomes” and it should be “exosomes”

Page 12, line 398 the word “currently” should be deleted. Then I will include a new phrase starting with “It is expected to achieve….” Then I will remove “in batch” as it does not add any information. In addition I will remove the words “in prospect” at the end of the manuscript because the absence of these words does not change the meaning of the phrase.

Comments on the Quality of English Language

As mentioned in the report, English is mostly correct but still contains several grammatical mistakes. Some phrases are too long. Some text can be removed without lost of meaning. The verb diagnose is used as sustantive (diagnostic should be used instead). Wrong use of analysis when analytical should be used. I included a few of these mistakes at the end of the comments but, it may be more to be spotted.

Author Response

(The authors gave the same response as above.)

Round 2

Reviewer 1 Report

Comments and Suggestions for Authors

The manuscript «Qualitative and quantitative analytical techniques of nucleic acid modification based on mass spectrometry for biomarker discovery» submitted by Ying Liu et al. to International Journal of Molecular Sciences is devoted to the review of approach to analysis of nucleic acid modification using mass spectrometry. Investigation of qualitative and quantitative composition od nucleic acid modification can be fruitful for development of new approach for diagnosis of different diseases such as cancer. The main problem of such studies is low content of nucleic acid modifications in biological samples. Authors discussed the variants of improvement of detection sensitivity and specificity and future perspectives of this direction.

The review is comprehensive and will be interesting for researcher from diverse scientific areas.

Comments on the Quality of English Language

Minor editing of English language required. The mistyping are still present in the text in single instance.

Reviewer 2 Report

Comments and Suggestions for Authors

The revised version has addressed all the suggestions explained in the first revision. In my opinion the manuscirpt can be accepted in the present form.